# Molecular Basis of Inhibitory Mechanism of Naltrexone and Its Metabolites through Structural and Energetic Analyses

**DOI:** 10.3390/molecules27154919

**Published:** 2022-08-02

**Authors:** Martiniano Bello

**Affiliations:** Laboratorio de Diseño y Desarrollo de Nuevos Fármacos e Innovación Biotecnológica, Escuela Superior de Medicina, Instituto Politécnico Nacional, Plan de San Luis y Diaz Mirón, s/n, Col. Casco de Santo Tomas, Mexico City 11340, Mexico; bellomartini@gmail.com or mbellor@ipn.mx

**Keywords:** naltrexone, mu-opioid receptor, MD simulations, MMGBSA, binding free energy

## Abstract

Naltrexone is a potent opioid antagonist with good blood–brain barrier permeability, targeting different endogenous opioid receptors, particularly the mu-opioid receptor (MOR). Therefore, it represents a promising candidate for drug development against drug addiction. However, the details of the molecular interactions of naltrexone and its derivatives with MOR are not fully understood, hindering ligand-based drug discovery. In the present study, taking advantage of the high-resolution X-ray crystal structure of the murine MOR (mMOR), we constructed a homology model of the human MOR (hMOR). A solvated phospholipid bilayer was built around the hMOR and submitted to microsecond (µs) molecular dynamics (MD) simulations to obtain an optimized hMOR model. Naltrexone and its derivatives were docked into the optimized hMOR model and submitted to µs MD simulations in an aqueous membrane system. The MD simulation results were submitted to the molecular mechanics–generalized Born surface area (MMGBSA) binding free energy calculations and principal component analysis. Our results revealed that naltrexone and its derivatives showed differences in protein–ligand interactions; however, they shared contacts with residues at TM2, TM3, H6, and TM7. The binding free energy and principal component analysis revealed the structural and energetic effects responsible for the higher potency of naltrexone compared to its derivatives.

## 1. Introduction

Naltrexone has been employed for decades as a therapy for alcoholism [1,2,3,4] and opiate dependency [5,6,7], with well-controlled clinical studies establishing it as an efficacious medication for the treatment of alcoholism [2,8,9]. A major disappointment of its employment has been that it is known to undergo a fast and extensive hepatic metabolism after administration. Naltrexone is a potent opioid antagonist, targeting endogenous opioid receptors (mu, delta, and kappa receptors), but particularly the mu-opioid receptor (MOR) [10,11]. MOR belongs to the rhodopsin family of G-protein coupled receptors (GPCRs), and it has been confirmed that the rhodopsin structure has a heptahelical domain on both sides of the plasma membrane [12,13,14,15]. Therefore, due to the lack of X-ray crystallography data for human MOR (hMOR), the seven transmembrane (TM) motifs of rhodopsin have been considered an appropriate model for other GPCRs [16,17]. However, the low sequence identity between opioid receptors and rhodopsin (30%) could be inconvenient for the construction of opioid receptor models based on the rhodopsin X-ray crystal structure through homology modeling.

Crystallographic studies have provided structural data on inactive, active, and fully active structures of murine MOR (mMOR) conformation. The tridimensional structure of inactive mMOR was co-crystallized with the morphine-like antagonist β-FNA [18]. This structure enabled the development of different theoretical studies to explore rational drug design [19,20] and structural changes associated with MOR activation [21,22]. More recently, the mMOR structure bound to the morphine-like agonist BU72 [23] and one stabilizing G protein mimetic camelid antibody fragment (Nb39) [24,25,26] was crystallized, providing more structural evidence about the activation of MOR, which is linked with its therapeutic function. This high-resolution mMOR structure and the high sequence identity (97%) with hMOR provide a unique chance to construct hMOR homology models for the evaluation of its binding between naltrexone and its derivatives.

Molecular dynamics (MD) simulation studies have provided relevant information about the MOR activation mechanism, taking the active or inactive MOR state as the initial conformer [23,27,28,29,30,31]. Based on these studies, we know that the active form of mMOR shows the following features compared with the inactive state: (1) a sizeable outward motion of TM6 relative to TM3; (2) minor inner motion of TM5 and TM7; (3) breakage of the hydrogen bond between Arg165 and Thr279 (equivalent to the Arg^3.50^−Asp/Glu^6.30^ GPCR ionic lock); and (4) the formation of one hydrogen bond between Arg165 and Tyr252.

The major metabolite of naltrexone in humans and other animal species is 6-β-naltrexol (NTXOL), and 2-hydroxy-3-O-methylnaltrexol (HMNTXOL) is another minor metabolite. NTXOL is also an opioid receptor agonist but is hundreds of times less potent than naltrexone at the mu-receptor [32]. Molecular modeling analysis based on molecular mechanics and semi-empirical calculations has shown that NTXOL and naltrexone share a comparable surface area and volume, indicating that both ligands may have a similar affinity for the same binding site in opioid receptors [33], contrasting with experimental finding [32]; therefore, it is important to explore the structural and energetic causes of this discrepancy.

In this study, we explore the structural and energetic basis of the mMOR in complex with naltrexone and its derivatives. First, hMOR was constructed through homology modeling methods by employing the high-resolution mMOR structure. The hMOR was anchored in a solvated phospholipid bilayer and submitted to one microsecond (µs) MD simulation to obtain an optimized hMOR model. Clustering analysis allowed the most populated hMOR conformer to be obtained, which was then used to perform docking studies and obtain hMOR–ligand complexes with naltrexone and its derivatives. The hMOR–ligand complexes were also built in an aqueous membrane system and submitted to µs MD simulations. These were followed by analyses relating to the binding free energies with the molecular mechanics–generalized Born surface area (MMGBSA) approach and principal component (PC) analysis.

## 2. Results

### 2.1. Docking Results

The docking studies showed the lowest binding free energy for the hMOR–naltrexone, hMOR–NTXOL, and hMOR–HMNTXOL complexes (Figure 1). For the hMOR–naltrexone complex, the ligand formed contacts with transmembrane helixes TM3, TM5, TM6, and TM7 through seven hydrophobic interactions (W135, I146, M153, V238, W295, I298, and V302), one charge residue (D149), and three polar residues (Y150, H299, and Y328). D149 in TM3 (equivalent to D147 in mMOR) also established two hydrogen bonds and one salt bridge with the protonated nitrogen atom of naltrexone (Figure 1A).

In the hMOR–NTXOL complex, NTXOL was coupled by six hydrophobic contacts: L58, L221, I298, V302, W320, and I324. Polar interactions were mediated by one charged and five polar residues: D149, T122, Q126, Y150, T220, and Y328 (Figure 1B). These interactions correspond to the extracellular region (ER) (L58), TM2 (T122 and Q126), TM3 (D149 and Y150), TM6 (I298 and V302), and TM7 (W320, I324, and Y328).

The hMOR–HMNTXOL complex is stabilized by seven hydrophobic residues: L58, W135, M153, T220, L221, W320, and I324. The polar contacts were formed by Q126, D149, Y150, and T220 (Figure 1C). These residues are placed along the ER (L58), TM2 (Q126), TM3 (D149, Y150, and M153), and TM7 (W320 and I324). From these polar residues, D149 and W320 formed hydrogen bonds with the ligand, whereas Y150 formed arene–H interactions. The hMOR–naltrexone, hMOR–NTXOL, and hMOR–HMNTXOL complexes were built in an aqueous membrane system and submitted to µs MD simulations to explore the molecular recognition under a dynamic and solvated environment.

### 2.2. Stability of the Systems

Before analyzing the results, we evaluated different geometrical parameters (the area per lipid headgroup, RMSD, and Rg) to observe whether the system was appropriately equilibrated. The area per lipid analysis allowed us to evaluate the bilayer thickness. Appendix A shows the time evolution of the area per lipid for the free and bound systems. This figure indicates that the systems exhibit higher area per lipid values at the beginning of the simulations, dropping off to a converged value within 0.15 µs, with average area per lipid values of 68.56, 69.56, 69.73, and 70.27 Å^2^ for the free hMOR, hMOR–naltrexone, hMOR–NTXOL, and hMOR–HMNTXOL systems, respectively, which are in line with the values reported for other protein–POPC–membrane systems [34]. Appendix A shows that free hMOR reached equilibrium between 0.3 and 0.4 µs, with average RMSD values of 10.0 Å. The hMOR–ligand systems reached equilibrium between 0.1 and 0.2 µs, with RMSD values of 2.9, 4.0, and 4.1 Å for the hMOR–naltrexone, hMOR–NTXOL, and hMOR–HMNTXOL systems, respectively (Appendix A). The significant difference in RMSD values between the free and bound hMOR systems is due to the hMOR–ligand systems were started from a pre-equilibrated hMOR conformer obtained over the equilibrated simulation of free hMOR system. Appendix A shows that the free and bound hMOR systems reached constant Rg values between 0.1 and 0.4 µs, with average values of about 25.5 Å. Based on these results, the first 0.4 µs were discarded from further analysis.

### 2.3. Ligand Interactions on the hMOR–Ligand Complex through MD Simulations

The most populated conformers through clustering analysis showed representative interactions of the three complexes. Naltrexone at the hMOR binding was bound by six hydrophobic residues at TM2 (A119), a loop connecting TM2-TM3 (W135), TM3 (M153), TM6 (W295 and I298), and TM7 (I324). The polar interactions were established by nine residues at TM2 (D116 and Q126), TM3 (D149 and N152), TM6 (H299), and TM7 (G327, Y328, N330, and S331). From these residues, D149 also formed one hydrogen bond and a salt bridge with the amine protonated atom of naltrexone (Figure 2A). Interactions involving D149 have also been reported for complexes between morphine, fentanyl, or naltrexone with hMOR in inactive, active, and full-active states [35].

The hMOR–NTXOL complex was coupled by eight hydrophobic residues at TM2 (L112, A115, and A119), TM3 (M153, I157, and L160), and TM6 (W295 and I298). It was also stabilized by seven polar interactions at TM2 (D116), TM3 (N152 and S156), and TM7 (G327, N330, S331, and N334). Of these, D116 forms a salt bridge with NTXOL (Figure 2B).

The hMOR–HMNTXOL complex is coordinated by eight hydrophobic residues at TM2 (A115 and A119), TM3 (M153), TM5 (V238), TM6 (W295, I298, and V302), and TM7 (I324). M153 also forms one hydrogen bond with naltrexone. The polar contacts were formed by residues at TM2 (D116), TM3 (D149, Y150, N152, and S156), TM6 (H299), and TM7 (G327, Y328, and S331) (Figure 2C). The comparative analysis of the most populated hMOR–naltrexone, hMOR–NTXOL, and hMOR–HMNTXOL complexes during MD simulations showed that the three complexes shared contact with nine residues: H2 (D116 and A119), H3 (N152, M153, and S156), H6 (W295 and I298), and H7 (G327 and S331). Comparison with docking results shows a higher number of residues in common among the three simulated complexes (Figure 1 and Figure 2).

### 2.4. Binding Free Energy Calculations

Binding free energy (ΔG_mmgbsa_) values were determined using the MMGBSA approach and were energetically favorable for the three hMOR–ligand complexes (Table 1). Table 1 also shows that the interaction energy (ΔE_MM_) was energetically more favorable for naltrexone than for NTXOL and HMNTXOL. The solvation free energy (ΔG_solv-GBSA_) contributed unfavorably to the ΔG_mmgbsa_ for the three complexes. The ΔG_solv-GBSA_ values suggest that naltrexone exhibits a higher desolvation cost than NTXOL and HMNTXOL, which contributes to the decrease in its ΔG_mmgbsa_ value. The ΔG_mmgbsa_ values were thermodynamically more favorable for naltrexone than for NTXOL and HMNTXOL, explaining the higher affinity of naltrexone.

### 2.5. Per-Residue Free Energy Decomposition for the hMOR–Naltrexone Complex

Appendix A displays the energies for each residue involved in the protein–ligand interactions of the hMOR–ligand systems. In the hMOR–naltrexone system, the major source of the binding free energy (ΔG_mmgbsa_ ≥ 1.0 Kcal) was M153, W295, I298, I324, G327, and Y328 (Table 2). Of these residues, G327 participates in forming hydrogen bonds through its polar backbone atoms (Figure 2A), M153, W295, I298, and I324 stabilize through hydrophobic interactions, and Y328 stabilizes by polar interactions. In the hMOR–NTXOL system, A115, A119, M153, I157, W295, N330, and N334 were the major contributors to ΔG_mmgbsa_. A115, A119, M153, I157, and W295 participate by forming hydrophobic contacts, and N330 and N334 by polar contacts (Figure 2B). For the hMOR– HMNTX system, M153, I298, I324, and Y328 mostly contributed to the binding affinity, of which M153 formed hydrophobic and one hydrogen bond with HMNTXOL, I298 and I324 formed hydrophobic contacts, and Y328 formed polar interactions (Figure 2C). A comparison of the ligand stabilization on the three systems indicates that naltrexone and NTXOL are better stabilized at the mu-receptor binding site than HMNTXOL, in line with the ΔG_mmgbsa_ values (Table 1). In the hMOR–naltrexone and hMOR–NTXOL systems, there were a similar number of residues contributing most to the affinity, but the residues were different; only M153 and W295 were shared in both complexes. In fact, the hMOR–naltrexone and hMOR–HMNTXOL systems shared contacts through a higher number of common residues (M153, I298, I324, and Y328) than the hMOR–NTXOL.

### 2.6. Principal Component (PC) Analysis

PC analysis identified the most important eigenvectors. For free and bound hMOR systems, the first two eigenvectors (PC1 and PC2) comprise the largest eigenvalues, containing 44.0, 69.0, 31.0, and 48.8% of the total mobility of free hMOR, hMOR–naltrexone, hMOR–NTXOL, and hMOR–HMNTXOL, respectively. Therefore, these eigenvectors have the main conformational states sampled during the simulations for the four systems (Figure 3). Projection onto the phase space of PC2 vs. PC1 shows that free hMOR (Figure 3A) covers a bigger region in the essential subspace than hMOR–naltrexone (Figure 3B), indicating that the latter has a lower conformational entropy compared to free hMOR. Figure 3C indicates that hMOR–NTXOL covers a significantly larger region in the essential subspace than free hMOR, suggesting a larger conformational entropy for hMOR–NTXOL. Figure 3D highlights that hMOR–HMNTXOL exhibits a similar distribution in the essential subspace to free hMOR, indicative of similar conformational behavior for these two systems. The diagonalized covariance matrix of backbone-heavy atoms demonstrated the following values: hMOR (29.2 nm^2^), hMOR–naltrexone (18.6 nm^2^), hMOR–NTXOL (41.5 nm^2^), and hMOR–HMNTXOL (26.2 nm^2^). These values indicate that the binding of naltrexone to hMOR contributes to decreasing the number of conformational states. In contrast, the binding of NTXOL increases the number of conformational states present in solution, whereas the binding of HMNTXOL to the hMOR binding site does not significantly impact the number of conformational states with respect to the free state.

The graphical representation of the total fluctuation along PC1 shows that the free and bound hMOR systems exhibit the highest collective motions along the extracellular domain, cytoplasmic domain, and the loop between TM5 and TM6 (Loop TM5-TM6) for the free hMOR (Figure 4A), hMOR–naltrexone (Figure 4B), and hMOR–NTXOL (Figure 4C) systems but only along the extracellular and cytoplasmic domains for hMOR–HMNTXOL (Figure 4D). hMOR–NTXOL exhibits the highest collective motions along the three abovementioned regions compared to free hMOR, hMOR–naltrexone, and hMOR–NTXOL (Figure 4C), systems, supporting the high heterogeneity observed through visualization onto the essential space (Figure 3C). For all the cases, the extracellular and cytoplasmic domains moved as a rigid body with respect to the transmembrane region containing the ligand binding domain. It is also appreciated that the reduction in conformational mobility of free hMOR–naltrexone compared to free hMOR took place along the extracellular domain, whereas similar mobility along the extracellular domain is observed for free hMOR and hMOR–HMNTXOL but in the opposite direction.

## 3. Discussion

Experimental studies by X-ray crystallography have shed insight into the active and inactive state of mMOR [18,23]. These structural data enabled the development of different theoretical studies, such as rational drug design and the exploration of the conformational changes associated with MOR activation [19,20,21,22,27,28,29,30,31]. However, there is currently no crystallographic information for hMOR. To obtain hMOR, homology modeling studies have been implemented, and these have been employed to explore structural changes present between the agonist or antagonist and hMOR [35,36,37]. In this study, we employed the high-resolution mMOR structure that has a high sequence identity (97%) with hMOR, along with its active state, to construct our hMOR homology model. Since hMOR is a membrane receptor, a complete membrane–aqueous system containing the modeled hMOR was built to optimize the receptor structure through 1 µs MD simulation. Based on cluster analysis over the equilibrated simulation time, the most populated hMOR conformer was selected to perform docking studies with naltrexone, NTXOL, or HMNTXOL. These hMOR–ligand complexes were also simulated in a membrane–aqueous environment through one µs MD simulations to relax and optimize binding interactions between the ligand and the amino acid residues in the hMOR binding cavity.

Docking studies showed that only four contacts were shared among the hMOR–naltrexone, hMOR–NTXOL, and hMOR–HMNTXOL complexes: I298, V302, D149, and Y150. Protein–ligand interactions involving D149 and Y150 have also been observed for complexes between hMOR in inactive, active, and full-active states with morphine, fentanyl, or naltrexone [35]. In fact, hMOR–NTXOL and hMOR–HMNTXOL shared a higher number of contacts (L58, Q126, Y150, D149, I298, V302, W320, and I324) than hMOR–naltrexone. 

We also observed differences in the type of residues stabilizing the three complexes after MD simulations; however, several residues present in the docking calculations were still present during MD simulations: residues at loop TM2-TM3 (W135), TM3 (D149, Y150, and M153), TM5 (V238), TM6 (W295, I298, H299, and V302), and TM7 (I324 and Y328). The salt bridge formed between D149 (D147 in mMOR) and the protonated nitrogen atom of ligand (Figure 1A), which was only observed for the hMOR–naltrexone complex, is an interaction previously inferred from experimental studies [38]. This interaction also favors interactions between TM3 and TM7 through one hydrogen bond between D149 and Y326 and MOR activation [38,39,40,41]. Interactions involving TM3 (M153), TM5 (V238), TM6 (H299 and V302), and TM7 (Y328) have also been reported for the hMOR–naltrexone complex through theoretical studies combining docking and MD simulations [35]. On the other hand, interactions with Y328 have been observed in MOR–agonist/antagonist complexes, and it has therefore been suggested that interactions with this residue could be associated with affinity rather than with efficacy [35]. Interactions with Y328 (Y326 in mMOR) may also be critical for the molecular recognition of naltrexone by hMOR, since residue mutation to phenylalanine decreases the affinity of naltrexone to mMOR [42].

ΔG_mmgbsa_ values determined using the MMGBSA method showed that the binding affinity was energetically more favorable for naltrexone than for its derivatives, explaining the higher potency of naltrexone compared with NTXOL, which is also a MOR antagonist but a hundred times less potent than naltrexone [32]. However, despite NTXOL showing a lower affinity, many other factors contribute to its lower potency, such as entropic effects, which we did not explore using the MMGBSA approach.

Per-residue decomposition analysis indicated that naltrexone and NTXOL are better stabilized at the MOR binding site than HMNTXOL, in line with the ΔG_mmgbsa_ values (Table 1). In the hMOR–naltrexone and hMOR–NTXOL systems, the quantity of residues that mainly contribute to the affinity were similar, but they were of a different type; only M153 and W295 were shared in both complexes. In fact, the hMOR–naltrexone and MOR–HMNTXOL systems shared more common contacts (M153, I298, I324, and Y328) than they shared with NTXOL.

PC analysis suggests that the favorable entropy contribution observed for the MOR–NTXOL complex could contribute to further decreasing the affinity predicted through the binding free energies reported in Table 1. In contrast, the unfavorable entropy observed for the hMOR–naltrexone complex may impact favorably on the reported binding affinity. Meanwhile, no important changes should be expected for the hMOR–HMNTXOL complex, since there are no important conformational changes with respect to free hMOR.

## 4. Material and Methods

### 4.1. Structural Modeling

The hMOR structure was constructed using homology modeling procedures. Modeler 9.17 [43] was used to build the hMOR model, using the high-resolution crystal structures of mMOR (PDB entry 5C1M_chainA) in the active state as a template, whose sequence (Uniprot, P42866) is 97.0% identical to human MOR (Uniprot, P35372). The best hMOR model was selected from this analysis based on the DOPE score of Modeler. Prediction of the transmembrane helices (Appendix A) in hMOR was carried out with the TMHMM 2.0 server [44]. The model was subjected to MolProbity analysis [45], reporting residues in favored regions of the Ramachandran plot. This structure was built in an aqueous membrane system and submitted to 1 µs MD simulations to optimize the internal interactions.

### 4.2. Docking Studies

The structures of naltrexone, NTXOL, and HMNTXOL were taken from ChemSpider (http://www.chemspider.com/ accessed on 20 August 2021) and optimized at the AM1 level with Gaussian 09W software [46]. The hMOR used to perform the docking calculations was obtained over the equilibrated simulation time (last 0.6 µs) through a clustering analysis of 1µs MD simulations (Appendix A). Docking studies were carried out using AutoDock Tools 1.5.6 and AutoDock 4.2.6, The Scripps Research Institute La Jolla, CA 92037 USA available online: https://autodock.scripps.edu/download-autodock4/ (accessed on 20 August 2021) [47]. In docking procedures, we used a rigid receptor structure and a flexible structure of the ligand. Hydrogen bonds and Gasteiger partial charges were assigned to ligands, and Kollman partial charges were placed on protein atoms. The Lamarckian genetic algorithm with an initial randomized population of 100 individuals and a maximum number of energy evaluations of 1 × 10^7^ was employed to generate binding poses. A grid box with a spacing of 0.375 and a size of 70 × 70 × 70 Å^3^ was constructed around the binding site to establish the ligand search location. The most populated cluster for the hMOR–ligand conformer, together with the lowest energy binding pose, was selected as the starting conformation for MD simulations. The docking protocol was validated by docking the co-crystallized ligand in mMOR (PDB entry 5C1M_chainA) in the most populated hMOR conformer obtained through MD simulations (Section 2.1). The root mean squared deviation (RMSD) of the docked pose with the lowest free energy was about 2.0 Å (Appendix A), indicating that the model remains near the initial conformation throughout MD simulation, supporting its use of reliable starting points to evaluate the ligand-binding impact.

### 4.3. Anchoring of the Receptor–Ligand Complex onto the Membrane

The orientation of the receptor–ligand complex with respect to the membrane was carried out using the OPM (Orientations of Proteins in Membranes) server [48]. A rectangular pre-equilibrated POPC (1-palmitoyl-2-oleoyl-sn-glycero-3-phosphocholine) membrane of dimensions 106.9 × 107.2 × 134.0 Å (xyz) was generated for each system using the membrane-builder tool of CHARM [49,50]. The replacement method was used to place the receptor–ligand complex into the POPC membrane, which comprised 290 POPC phospholipids. The protein–receptor–membrane system was solvated with 31068 TIP3 water molecules and neutralized with 0.15 M NaCl using the ion-placing method.

### 4.4. MD Simulations

The protein–receptor–membrane systems were submitted to MD simulations using the Amber 16 package [51]. Ligand parameters were obtained using the generalized AMBER force field (GAFF), considering the AM1-BCC method and GAFF to assign atomic charges. Topologies for the systems were constructed with a Leap module using ff14SB [52], Lipid14 [53], and GAFF [54]. The systems were energy-minimized with position restraints on the protein–receptor–membrane atoms, allowing relaxation of the solvent. Systems were gradually heated from 0–310 K for one nanosecond (ns) under the NVT ensemble, with the restraint of the heavy atoms of the protein–receptor–membrane system. The system was equilibrated for one ns under the NPT ensemble at 310 K and 1 bar pressure with the restrained heavy atoms, followed by five ns with the entirely unrestricted system. Triplicate MD simulations were run for one µs for each system under periodic boundary conditions (PBCs) using an NPT ensemble at 310 K and 1 bar pressure. Long-range electrostatic interactions were treated with the particle mesh Ewald [55], considering a 10 Å cutoff for van der Waals interactions. The SHAKE algorithm [56] was used to restrict bond lengths at the equilibrium. Pressure was maintained using a semi-isotropic constant surface tension to preserve the area per lipid. The temperature was maintained using Langevin dynamics.

### 4.5. MD Trajectory Analysis

The area per lipid headgroup, the time-dependent Cα RMSD, the radius of gyration (Rg), and clustering analysis were estimated using AmberTools16. The equilibrated part of each simulation was concatenated into a single joined trajectory and employed to evaluate the PC analysis, clustering analysis, and binding free energy analysis. PC analysis was calculated from the diagonalization of the covariance matrix of backbone atoms [57]. From this analysis we obtained a set of eigenvalues related to eigenvectors. From these, the first two eigenvectors (PC1 vs. PC2) were considered as reaction coordinates to construct the projection of the systems in phase space. Figures were created using PyMOL [58].

### 4.6. Binding Free Energy

The binding free energies of the receptor–ligand interactions were calculated using the MMGBSA method [59]. The binding free energy was estimated over the equilibrated simulation time, saving 4000 receptor–ligand conformations. The solvation free energy was determined using implicit solvent models [60] and an ionic strength of 0.15 M. The ΔG_bind_ values using the MMGBSA approach were estimated as reported elsewhere [61].

## 5. Conclusions

High resolution crystallographic structure data, homology modeling, docking, and MD simulations analyses were used to explore the thermodynamics and dynamic analysis of the molecular recognition of naltrexone and its derivatives with MOR. Clustering analysis showed that naltrexone and its derivatives were stabilized by residues at loop TM2-TM3, TM3, TM5, TM6, and TM7. However, the characteristic salt bridge between D149 at TM3 and the protonated nitrogen atom of naltrexone, which favors interactions between TM3 and TM7 and subsequent MOR activation, was only present for the hMOR–naltrexone complex. Thermodynamic and clustering analysis showed that naltrexone and its derivatives were bound at the hydrophobic cavity of MOR, with the binding free energy order of naltrexone ≥ NTXOL ≥ HMNTXOL, which correlates with the higher potency of naltrexone compared with NTXOL. Although structural analysis demonstrated that naltrexone and its derivatives shared contacts with a group of similar residues, per-residue decomposition analysis indicated that a major source of the affinity came from different residues for the hMOR–naltrexone and hMOR–HMNTX complexes. Finally, PC analysis revealed that the unfavorable entropy contribution observed for the hMOR–naltrexone complex may contribute to improving the binding affinity of naltrexone for hMOR, contrasting with the hMOR–NTXOL or hMOR–HMNTXOL complexes, where favorable or no conformational changes were observed in the molecular recognition for hMOR–NTXOL or hMOR–HMNTXOL, respectively.

## Figures and Tables

**Figure 1 molecules-27-04919-f001:**
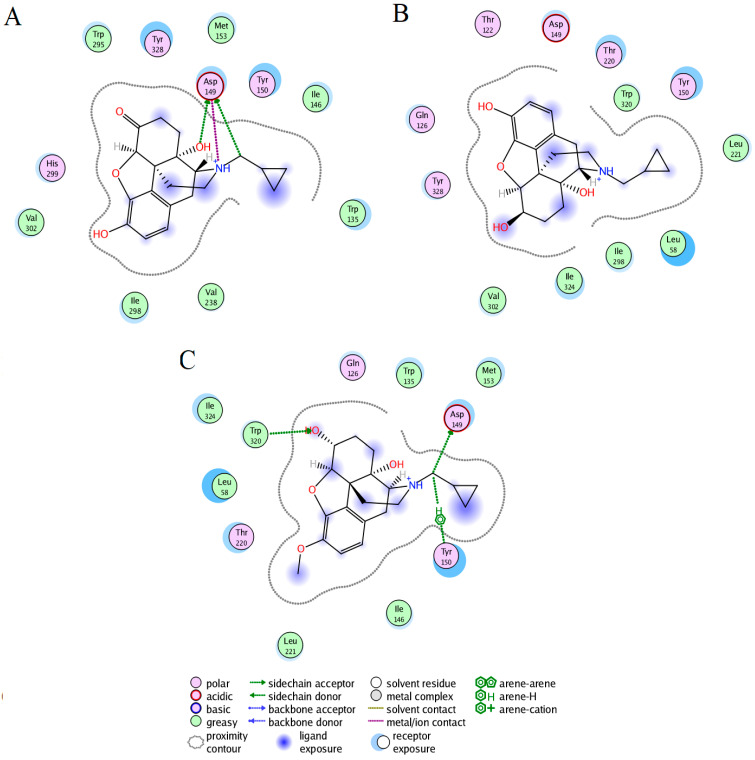
Docking interactions of naltrexone, NTXOL, or HMNTXOL with hMOR. Interactions of naltrexone (**A**), NTXOL (**B**), or HMNTXOL (**C**) with hMOR.

**Figure 2 molecules-27-04919-f002:**
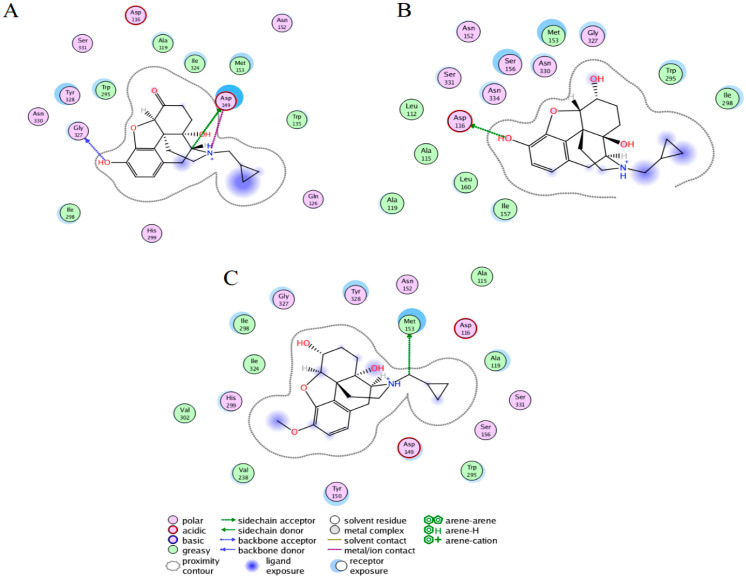
Protein–ligand interactions of naltrexone, NTXOL, or HMNTXOL with hMOR. Interactions of naltrexone (**A**), NTXOL (**B**), or HMNTXOL (**C**) with hMOR present in the most populated conformations through MD simulations.

**Figure 3 molecules-27-04919-f003:**
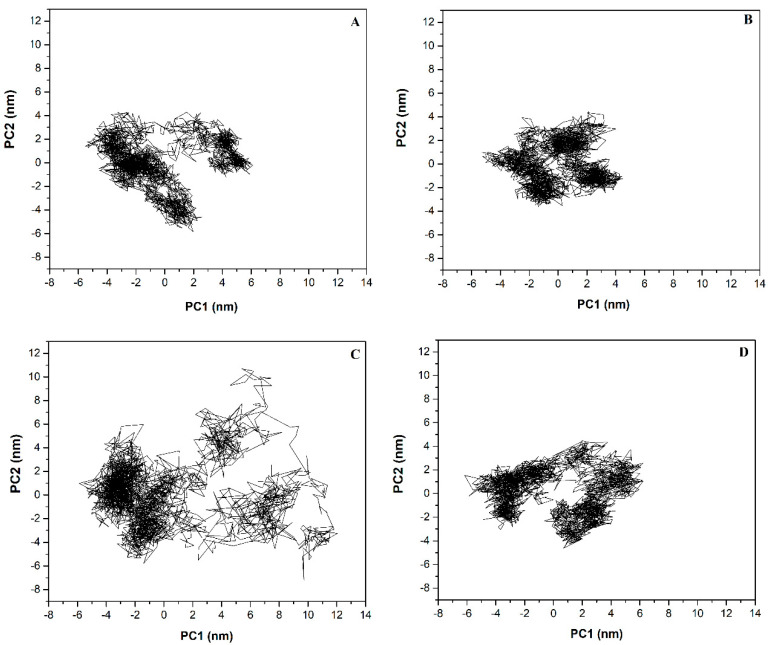
Projection of the free and bound hMOR systems in phase space. Projection of the motion in the phase space along PC2 vs. PC1 for free hMOR (**A**), hMOR-naltrexone (**B**), hMOR-NTXOL (**C**), and hMOR-HMNTXOL (**D**).

**Figure 4 molecules-27-04919-f004:**
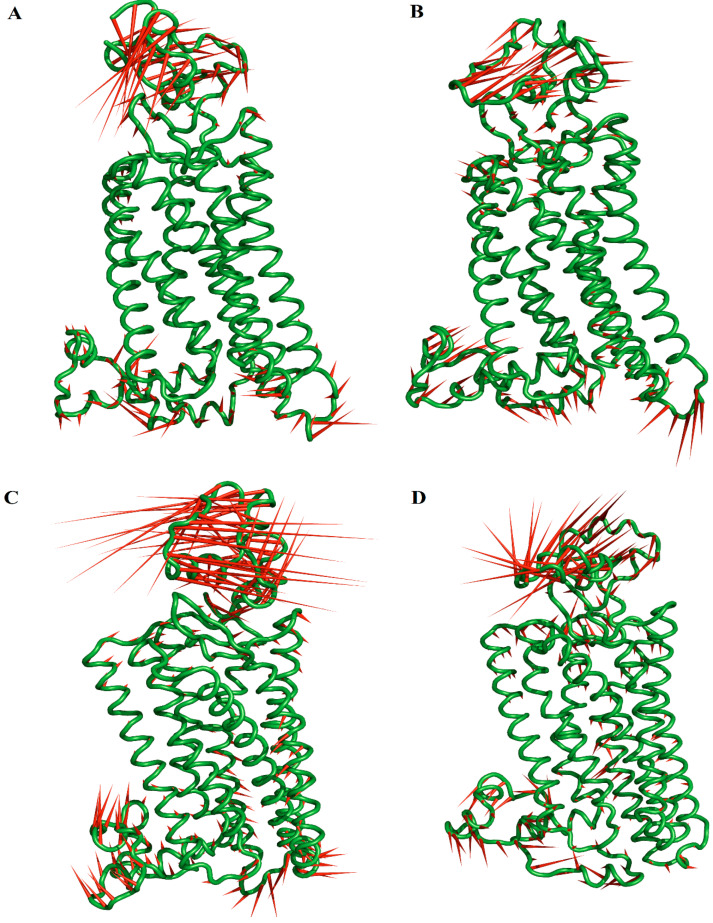
Graphic representation of the two extreme projections of the free and bound hMOR systems. Graphical depiction of the two extreme projections along PC1 vs. PC2 for the free hMOR (**A**), hMOR–naltrexone (**B**), hMOR–NTXOL (**C**), and hMOR–HMNTXOL (**D**) systems. The direction and magnitude of motions are depicted as porcupine depictions.

**Table 1 molecules-27-04919-t001:** Binding free energy components for protein–ligand interactions of hMOR–naltrexone, hMOR–NTXOL, and hMOR–HMNTXOL systems calculated using the MMGBSA approach (values in kcal/mol).

Systems	ΔE_MM_	ΔG_solv-GBSA_	ΔG_MMGBSA_
hMOR–naltrexone	−71.36 ± 7.12	37.97 ± 6.11	−33.39 ± 2.68
hMOR–NTXOL	−39.43 ±11.8	8.76 ± 1.0	−30.67 ± 2.95
hMOR–HMNTXOL	−49.99 ± 8.5	20.36 ± 8.7	29.63 ± 2.62

**Table 2 molecules-27-04919-t002:** Per-residue free energy for hMOR–ligand interaction complexes (values in kcal/mol).

Residue	hMOR–Naltrexone	hMOR–NTXOL	hMOR–HMNTXOL
A115		−1.173	
A119		−1.128	
M153	−1.780	−1.551	−3.050
I157		−1.016	
W295	−1.572	−1.176	
I298	−1.184		−1.725
I324	−1.216		−1.382
G327	−1.547		
Y328	−2.774		−1.522
N330		−1.566	
N334		−1.942	

## Data Availability

The datasets supporting the conclusions of this research are contained within the paper and its Appendix A.

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
