# Peer review of "Molecular Basis of Inhibitory Mechanism of Naltrexone and Its Metabolites through Structural and Energetic Analyses"

_molecules, 2022, doi:10.3390/molecules27154919_

Round 1

Reviewer 1 Report

The manuscript describes theoretical work which has been executed to a high standard and is in a highly topical area so I expect it to be of significant general interest. It is generally very well written and the methods describe the techniques used comprehensively. The following fairly minor points may help the authors revise the manuscript. 

Lines 39 to 49. A couple of times in this paragraph it is states that a 'crystal structure' was 'crystallized' which is slightly odd because the protein is crystallised, not the crystal structure! It might be better to say that the crystal structures were 'determined'. 

Lines 57 to 63. Here it is stated that NTXOL is hundreds of times less potent, but later it says that it has comparable affinity with NTX. This needs to be clarified. Also, strictly speaking, they are ligands rather than substrates?

There are large differences between the 4 structures shown in Fig. 4 affecting two of the three domains of the receptor. Firstly, it would be interesting to indicate which of these domains are involved in ligand binding and then to describe the structural differences more. For example, are they rigid-body movements of the smaller domains or are there local structural changes within these domains, or both?

Author Response

Response to reviewers

I want to thank reviewers for their fruitful comments and observations that, without hesitation, will contribute to improving the quality of this scientific contribution.

The manuscript describes theoretical work which has been executed to a high standard and is in a highly topical area so I expect it to be of significant general interest. It is generally very well written and the methods describe the techniques used comprehensively. The following fairly minor points may help the authors revise the manuscript. 

Lines 39 to 49. A couple of times in this paragraph it is states that a 'crystal structure' was 'crystallized' which is slightly odd because the protein is crystallised, not the crystal structure! It might be better to say that the crystal structures were 'determined'. 

Response:

The mistake has been corrected.

Lines 57 to 63. Here it is stated that NTXOL is hundreds of times less potent, but later it says that it has comparable affinity with NTX. This needs to be clarified. Also, strictly speaking, they are ligands rather than substrates?

Response:

We have added more information to clarify this phrase. Although theoretical studies suggest a similar affinity for the same binding site in opioid receptors, this contrast with experimental finding [32], therefore, it is important to explore the structural and energetic causes of this discrepancy. This information has been added in the manuscript.

There are large differences between the 4 structures shown in Fig. 4 affecting two of the three domains of the receptor. Firstly, it would be interesting to indicate which of these domains are involved in ligand binding and then to describe the structural differences more. For example, are they rigid-body movements of the smaller domains or are there local structural changes within these domains, or both?

Response:

For all the cases, the extracellular and cytoplasmic domains moved as a rigid body with respect to the transmembrane region containing the ligand binding domain. This information has been included in the manuscript.

Reviewer 2 Report

The manuscript molecules-1808433 "Molecular basis of inhibitory mechanism of naltrexone and its metabolites through structural and energetic analyses" by Martiniano Bello is correctly organized and brings new scientific results that deserve publication in the Molecules journal after the corrections/improvements described below.

1)Line 19 „“they shared contact with residues at TM2, TM3, H6” – should it be ‘contacts’?

2)In the sentence in lines 180-183 and part 2.6 - it is necessary to explain in detail which input data the eigenvectors were constructed (derived).

3)Explain in a little more detail whether you used a flexible local or rigid structure of the protein and a flexible or rigid structure of the ligand in the docking procedure.

4)Describe which location markers in the transmembrane (TM) segment sequence did you use? If an algorithm was used to predict the position of TM segments in the sequence - please specify which one. It would be useful to provide information about the sequence and position of TM segments in supplementary information or the main text (Figure, Table). 

5)The part of the sentence in lines 223-224, “Homology modelling studies have therefore been implemented to obtain hMOR,…. “ must be improved. It should be “to obtain the 3D structure of hMOR…”.  It is necessary to be precise in expression/description - here and in other parts of the manuscript.

6)In my opinion, it is not good, and it is not customary in the chemical literature, to abbreviate the name of a compound with a short name like Naltrexone (NTX) and use it in all parts of the manuscript. There is no need for that here. The problem is that it is not a common abbreviation, and the average reader will not be able to understand parts of the manuscript by starting to read from the figures/tables or some later part - they will have to look up the meanings of the abbreviations first.

7)It would be useful to provide 3D structural information as a PDB file for the most important docking poses.

8)It is good if at least some parts of the results obtained by modelling can be connected and possibly confirmed with some measured/experimental data from the literature. If this is possible in the case of these results, it would be useful for the author to do so - or to comment/explain.

Author Response

Reviewer 2.

I want to thank reviewers for their fruitful comments and observations that, without hesitation, will contribute to improving the quality of this scientific contribution.

he manuscript molecules-1808433 "Molecular basis of inhibitory mechanism of naltrexone and its metabolites through structural and energetic analyses" by Martiniano Bello is correctly organized and brings new scientific results that deserve publication in the Molecules journal after the corrections/improvements described below.

1)Line 19 „“they shared contact with residues at TM2, TM3, H6” – should it be ‘contacts’?

Response:

The mistake has been corrected.

2)In the sentence in lines 180-183 and part 2.6 - it is necessary to explain in detail which input data the eigenvectors were constructed (derived).

Response:

The information about input data has been detailed at material and methods (section 4.5).

3)Explain in a little more detail whether you used a flexible local or rigid structure of the protein and a flexible or rigid structure of the ligand in the docking procedure.

Response:

The suggested information has been added in the manuscript (section 4.2)

4)Describe which location markers in the transmembrane (TM) segment sequence did you use? If an algorithm was used to predict the position of TM segments in the sequence - please specify which one. It would be useful to provide information about the sequence and position of TM segments in supplementary information or the main text (Figure, Table). 

5)Response:

TMHMM 2.0 was used to predict the transmembrane helices in MOR. This information has been added in the manuscript as supplementary material.

The part of the sentence in lines 223-224, “Homology modelling studies have therefore been implemented to obtain hMOR,…. “ must be improved. It should be “to obtain the 3D structure of hMOR…”.  It is necessary to be precise in expression/description - here and in other parts of the manuscript.

Response:

The suggestion has been considered in the manuscript.

6)In my opinion, it is not good, and it is not customary in the chemical literature, to abbreviate the name of a compound with a short name like Naltrexone (NTX) and use it in all parts of the manuscript. There is no need for that here. The problem is that it is not a common abbreviation, and the average reader will not be able to understand parts of the manuscript by starting to read from the figures/tables or some later part - they will have to look up the meanings of the abbreviations first.

Response:

The suggestion has been considered in the manuscript.

7)It would be useful to provide 3D structural information as a PDB file for the most important docking poses.

Response:

The suggestion information has been included in the manuscript as supplementary material.

8)It is good if at least some parts of the results obtained by modelling can be connected and possibly confirmed with some measured/experimental data from the literature. If this is possible in the case of these results, it would be useful for the author to do so - or to comment/explain.

Response:

We have considered this suggestion in the manuscript.